# Effectiveness of mHealth Interventions for Monitoring Antenatal Care among Pregnant Women in Low- and Middle-Income Countries: A Systematic Review and Meta-Analysis

**DOI:** 10.3390/healthcare11192635

**Published:** 2023-09-27

**Authors:** Manisha Mishra, Debasini Parida, Jogesh Murmu, Damini Singh, Tanveer Rehman, Jaya Singh Kshatri, Sanghamitra Pati

**Affiliations:** ICMR-Regional Medical Research Centre, Bhubaneswar 751023, India; mishram545@gmail.com (M.M.); debasiniparida96@gmail.com (D.P.); mjogesh14@gmail.com (J.M.); daminis274@gmail.com (D.S.); drtanveerrehman@gmail.com (T.R.)

**Keywords:** mHealth, pregnant women, antenatal care, effectiveness

## Abstract

Antenatal care (ANC) is essential in maternal and child health since it provides care to pregnant women from conception through to labour in order to ensure a safe pregnancy and childbirth. In recent years, mobile health (mHealth) interventions have emerged as a promising solution to improve maternal and child health outcomes in low- and middle-income countries (LMICs). The present study aimed to conduct a systematic review and meta-analysis of trials to evaluate the effectiveness of mHealth interventions to monitor prenatal care among pregnant women in LMICs. A systematic literature review was conducted using the databases CINHAL, Embase, MEDLINE, and PsycINFO on the effectiveness of mHealth interventions in monitoring the antenatal care of pregnant women. The study selection, data extraction of the included articles, and quality appraisal were assessed. Our study included six studies considering 7886 participants. All articles were from low- and middle-income countries (LMICs). Antenatal mothers who used a mobile health intervention were more likely (RR = 1.66, 95%CI = 1.07–2.58, I^2^ = 98%) to attend ANC check-ups when compared with the women who did not use any mobile health applications or did not receive any short message services. mHealth technologies are being utilised more and more to increase care accessibility and improve maternal and fetal health. Policymakers should prioritise the integration of mHealth interventions into maternal healthcare services in LMICs, ensuring that they are cost-effective, accessible, and sustainable and that healthcare workers are trained to deliver these interventions effectively.

## 1. Introduction

Despite progress in reducing global maternal mortality, it remains unacceptably high, particularly in low- and middle-income countries (LMICs) [1]. LMICs are defined as those with a GNI per capita between USD 1036 and USD 4045 according to the World Bank. Efforts towards achieving Sustainable Development Goal (SDG) 3.1, aimed at reducing maternal mortality, have fallen short of expectations due to many factors. These include inadequate quality of care, insufficient access to family planning, disparities in socioeconomic status and race/ethnicity, inadequate allocation of national resources, and inadequate health system infrastructure [2]. Antenatal care (ANC) is critical in maternal and child health by providing care to pregnant women from conception until labour to ensure a healthy pregnancy and childbirth. The World Health Organization (WHO) recommends that pregnant women receive at least eight antenatal care contacts, which include a range of services such as physical examinations, laboratory tests, health education, and counselling [3,4]. Antenatal care (ANC) is beneficial in promoting full-term births with average birth weight [5].

Consequently, there is an immediate requirement for innovative and efficient measures to enhance the well-being and survival of both mothers and infants [6]. According to the WHO, 94% of maternal deaths occur in LMICs, with sub-Saharan Africa and South Asia accounting for most of these deaths [7]. In LMICs, only 66% of pregnant women receive the recommended four or more antenatal care visits [8]. In India, which has one of the highest maternal mortality rates in the world, only 51% of pregnant women receive the recommended number of antenatal care visits. The lack of access to adequate antenatal care contributes to maternal and infant mortality and morbidity in LMICs [9], with complications such as pre-eclampsia, postpartum haemorrhage (PPH), and sepsis leading to maternal deaths [10].

Skilled care before and after childbirth can lower complications and prevent maternal deaths [11,12]. The government has implemented various programs like Janani Suraksha Yojana (JSY), Janani Shishu Suraksha Karyakram (JSSK), Pradhan Mantri Surakshit Matritva Abhiyan (PMSMA), and LaQshya to improve maternal and child health care. Various IEC materials, such as training packages, manuals, booklets, and videos, have been created to enhance the effectiveness of these programs, and capacity-building programs like Skilled Attendance at Birth and DAKSHATA have also been implemented [13]. In recent years mobile health (mHealth) interventions have emerged as a promising solution to improve maternal and child health outcomes in LMICs [14]. mHealth solutions offer a convenient and affordable way to overcome the challenges of accessing healthcare in LMICs. They can benefit pregnant women in remote and rural areas by enabling online/telephonic consultations, referrals, and appointment scheduling with community health workers. Furthermore, mHealth tools can aid in data collection and maintaining accurate health records [15,16].

Several mHealth interventions have been developed for antenatal care in LMICs, which use mobile devices to send reminders, educate pregnant women, and monitor their health status [17,18]. Studies show that text messaging and mobile phone-based interventions have successfully enhanced access to antenatal care and improved maternal and neonatal health outcomes in LMICs, reducing the incidence of stillbirths and neonatal mortality [19,20].

Mobile health (mHealth) interventions have been proposed to improve antenatal care access and quality in LMICs. The mHealth interventions used are short message service (SMS), voice messaging, notification alerts through a mobile application, and IVRS (interactive voice response system). Some trials have been conducted in countries among pregnant women regarding their development of knowledge through mobile health technologies to improve their prenatal healthcare. Therefore, to make noticeable policy changes with a compilation of individual studies, the present study aimed to conduct a systematic review and meta-analysis of trials to evaluate the effectiveness of mHealth interventions to monitor prenatal care among pregnant women in LMICs.

## 2. Materials and Methods

### 2.1. Criteria for Considering Studies for the Review

Types of studies—All published randomised, quasi-experimental, and cluster randomised trials were included in this analysis. After making an unsuccessful effort to contact the study author for more information, studies reported in abstract form, without adequate information on study methods, or where the findings were ambiguous were excluded. This systematic analysis excluded studies involving case reports, case studies, editorials, perspectives, literature reviews, conference abstracts, observational studies, commentaries, and studies published in languages other than English.

The inclusion criteria were peer-reviewed articles, published in the English language, published studies conducted in LMICs, published randomised, quasi-experimental, and cluster randomised trials.

The exclusion criteria were observational studies, literature reviews, case reports, meta-analyses, and systematic reviews.
b.Types of participants—Pregnant women (from the first trimester to the third trimester) (up to delivery) in LMICs.
-The pregnant women were followed by routine care with mHealth intervention and utilized a mobile application.c.Types of interventions—mHealth application for monitoring antenatal health intervention activities—short message service (SMS), voice calling, voice messaging, notification alerts through a mobile application, and IVRS (interactive voice response system).d.Types of outcome measures
Primary outcomes for the mother: ANC attendance

### 2.2. Search Methods for the Identification of Studies

Data sources and search strategy—This study was conducted according to the Preferred Reporting Items for Systematic Reviews and Meta-analyses (PRISMA) reporting guidelines [21]. We comprehensively searched the MEDLINE, Embase, and CINAHL databases from inception from January to May 2022. Additional citations were sought from the references in articles retrieved through searches.

### 2.3. Data Collection and Analysis

Selection of studies—The studies were selected independently by two reviewers (MM and DP) by reading their titles and abstracts through the use of Rayyan software, and disagreements were resolved through discussion with a third reviewer (JM). Full articles were retrieved for studies that did not provide a summary or had a scant abstract to determine their eligibility for inclusion. Two review authors (MM and DP) independently evaluated the full texts of the remaining papers for inclusion. Disagreements between them were solved through discussion and concord. A third review author was involved in deciding if no conclusion was reached. Details of the screening are provided below in the PRISMA flow chart.Data extraction and management—Using a predesigned data extraction form, two review authors individually retrieved the following information: (1) study information (study ID, date of extraction, title, authors, and source of study if not published); (2) study characteristics (study design, sample size, and inclusion/exclusion criteria used in the study, geographical location, and setting); (3) characteristics of the participants, including population type and mean age; (4) details of interventions; and (5) outcomes as described in the outcome measures above. Assessment of risk of bias in the included studies—We used the ‘Revman risk of bias’ assessment tool to assess the risk of bias for the included studies. Two review authors (JM and DS) independently evaluated the risk of bias in the included studies—the details are presented in Table 1 (risk of bias table). Disagreements were resolved through discussion or a conversation with a third review author. Measurement of treatment effect—The statistical analysis was executed according to [22]. We expressed effect measures as risk ratios and risk differences with 95% confidence intervals in the case of dichotomous outcomes. Based on the numerator and denominator of the studies where the risk ratio was not reported, we calculated the risk ratio between mothers who used a mobile phone and those who did not in both the intervention and control arms. Assessment of reporting bias—By determining whether the study was registered in a trial registry, a protocol was accessible, and an outcome was provided in the methods section, reporting bias was evaluated. The list of results from those sources was contrasted with the results mentioned in the paper that was published. An inverted funnel plot was used to assess potential publication bias. Data synthesis—We used Review Manager 5 software (version 5.4) to carry out the meta-analysis for this study. The meta-analysis was carried out by statistically combining the pregnancy outcomes of increased attendance at antenatal check-ups in various studies. The statistical analyses were carried out according to [22]. We conducted a random-effect meta-analysis where appropriate. We used funnel plots to assess publication bias in the meta-analysis with I^2^ value and heterogeneity.Quality of evidence—A quality check of the evidence was carried out by Review Manager 5 software (version 5.4) using the risk of bias tool depicting the included studies at high, low, or moderate risk. In all analyses, a *p*-value of less than 0.05 was considered statistically significant.

## 3. Results

### 3.1. Identification and Characteristics of Included Studies for SR and MA

A comprehensive electronic search of various databases revealed a total of 541 articles. After removing 73 duplicate articles, we scrutinised the titles and abstracts of 468 articles, out of which 46 were full-text articles. However, as illustrated in the PRISMA flow diagram (Figure 1), we excluded 40 articles for various reasons like different primary outcomes for some studies, different methods of study, and those studies which did not fulfill our eligibility criteria. A total of six studies, those that ideally addressed the PICO question for systematic review and meta-analysis, were included with a total of 7886 women. 

P—pregnant women (from the first trimester to the third trimester) (up-to delivery) in India. I—mHealth application for monitoring antenatal health. C—pregnant women not using mobile health apps. O—decreased maternal mortality. 

An overview of the characteristics of each study, which targeted antenatal women in the first to the third trimester in rural and remote areas of low- or lower-middle-income countries, and all of the RCTs performed in healthcare facilities including in India (one study conducted at the Rural Medical College, Loni, Ahmednagar), Zanzibar (two studies), Nigeria (one study), Brazil (one study) and another study conducted in Ethiopia are provided in Table 1.

All of the studies compared antenatal mothers with a mobile health intervention to a control group of antenatal mothers. Out of the six studies, the “M SAKSHI” application was used as an intervention for one study [25], whereas in another five studies, short message services, text messages, and mobile phone calls were the mode of intervention for monitoring the outcome effect (antenatal health) (Table 2).

### 3.2. Results for ANC Attendance

Antenatal mothers who used a mobile health intervention were more likely (RR = 1.66, 95% CI = 1.07–2.58, I^2^ = 98%) to attend ANC check-ups when compared with the women who did not use any mobile health applications or did not receive any short message services (Figure 2). 

A funnel plot (Figure 3) using data from the six reported studies representing asymmetrical distribution, which signifies a chance of publication bias.

## 4. Discussion

The primary goal of conducting a systematic review and meta-analysis on the effectiveness of mHealth interventions for monitoring antenatal health is to develop or strengthen the evidence for the effectiveness of mHealth in antenatal care. Moreover, this study will aid in determining the potential to significantly improve ANC attendance among antenatal mothers, thereby contributing to improved maternal and fetal health outcomes in low-income countries. Several studies have addressed the use of mHealth to promote MNCH in LMICs, but only a small number have thoroughly assessed how these interventions have affected health outcomes in these populations. mHealth technologies are being utilised more and more to increase care accessibility and improve maternal and fetal health. Unfortunately, compared to women in higher-income countries, women in low-income countries have a 120-fold higher risk of dying from causes related to pregnancy and childbirth [29]. Therefore, in the present meta-analysis, we pooled data from selected studies conducted in low- and middle-income countries. 

In the present study, we found that antenatal mothers who used mobile health interventions had an approximately two times higher chance of receiving ANC check-ups than women who did not use any mobile health applications or did not receive any short message services, which is higher than in another systematic review conducted in a population of LMICs in the time frame from 2008–2020 (OR = 1.89, 95% CI: 1.49–2.19) [30]. Despite the fact that most people exhibit techno-skepticism (i.e., a skeptical attitude toward technology), this meta-analysis revealed that mHealth has the potential to increase antenatal attendance when compared to conventional approaches. A further randomised controlled trial was carried out in Zanzibar between March 2009 and March 2010 in low- and middle-income nations, which confirmed these results. It showed that women who received a mobile phone intervention were more than twice as likely to receive four or more antenatal care visits compared to those who did not receive the intervention [23]. Irrespective of the typically low antenatal coverage in Sub-Saharan African countries, our findings were supported by a prospective randomised controlled trial conducted in Kenya in 2012 [31]. However, another systematic review and meta-analysis carried out in Ethiopia discovered that women who were sent text messages were around three times more likely to attend ANC appointments compared to those who did not receive such messages (OR = 2.74 95% CI: 1.41, 5.32) [15], which shows a greater result compared to our study.

Technology does not discriminate, and a well-functioning system has the potential to improve service coverage equality for both disadvantaged and non-marginalised populations [32]. Similarly, a USAID-supported study in Afghanistan resulted in not only mass participation but also enhanced the health-seeking behavior of the respective community [33]. Thus, it is important to comprehend the trend and pattern of mobile ownership across different geographic scales. Pregnant women who received text messages reported higher levels of satisfaction than women who received standard prenatal care, according to Jareethum et al.’s optimistic data from Thailand. Similarly, Kaewkungwal et al. [34] demonstrated that using mHealth in Thailand increased the coverage of ANC and immunization among expectant mothers and kids. This implies that other resource-poor nations like India may use mobile text messages as a promising behavior-change communication approach [35]. Clinical mHealth cannot be successful on its alone without the use of such cost-effective technologies in public health. Healthcare systems may develop a comprehensive and sustainable ecosystem that utilises the benefits of both areas by emphasizing collaboration and integration of clinical mHealth with cost-effective technologies in public health. This integration maximises clinical mHealth’s potential to improve healthcare delivery, improve public health outcomes, and contribute to individuals’ and communities’ overall well-being.

### 4.1. Policy Implications

Integration of mHealth interventions: Policymakers should prioritise the integration of mHealth interventions into the existing maternal health care services in LMICs. This will ensure that antenatal mothers have access to mHealth interventions and traditional healthcare services [36,37].Cost-effective interventions: When compared to traditional healthcare services, mHealth interventions have the potential to improve maternal health outcomes at a lower cost. When making decisions about maternal health care, policymakers should consider the cost-effectiveness of these interventions.Accessibility of technology: Policymakers should prioritise increasing access to mobile phones and Internet services, as these are essential for mHealth interventions to be effective. This may involve increasing investment in telecommunication infrastructure and making Internet and mobile phone services more affordable [38].Capacity building: India has launched various initiatives to improve public healthcare infrastructure, including the National Rural Health Mission (NRHM) and the Indian Public Health Standards (IPHS). Structural changes have been proposed to provide quality care to rural populations, and training and capacity building of healthcare personnel is critical. Policymakers must prioritise building the capacity of healthcare workers to deliver effective mHealth interventions, including training on the use of mHealth tools and ensuring access to necessary resources [39].Monitoring and evaluation: To ensure the effectiveness and sustainability of mHealth interventions for maternal health outcomes, policymakers should prioritise monitoring and evaluating their impact. In resource-limited settings, mHealth interventions should be considered by public health practitioners, policymakers, and researchers. This study’s findings and evidence-based recommendations could be useful in addressing maternal healthcare challenges in low- and middle-income countries (LMICs) through the use of various mHealth interventions, thereby contributing to the achievement of the Sustainable Development Goal of Maternal and Child Health [40,41].

Overall, policymakers should prioritise the integration of mHealth interventions into maternal healthcare services in LMICs, ensuring that they are cost-effective, accessible, and sustainable and that healthcare workers are trained to deliver these interventions effectively.

### 4.2. Strengths and Limitations of This Study

Even with limited support such as SMS, there has been an improvement in the utilization of antenatal services amidst the changing landscape of technology.

The inclusion of studies from low- and middle-income countries (LMICs) enables results to be generalised to their respective populations.

As an intervention, the broad term “mHealth” has resulted in heterogeneity.

The limitation of this study is that only articles published in English were included, which may have resulted in the exclusion of relevant studies published in other languages. However, this decision was made to ensure consistency in the intervention, study design, participants, and outcome measures.

## 5. Conclusions

The emergence of mobile health technology (mHealth) has made it possible to improve prenatal care and empower pregnant women. It enhances emergency obstetric referrals, promotes collaboration among health workers, and improves overall care delivery. Moreover, mHealth can also strengthen preventative services through the widespread dissemination of antenatal care promotion. Finally, mHealth has revolutionised prenatal care by making it more accessible and providing pregnant women with the knowledge and resources they need to actively participate in their own prenatal care. This not only improves individual results but also benefits community health and well-being because healthier pregnancies result in healthier populations.

## Figures and Tables

**Figure 1 healthcare-11-02635-f001:**
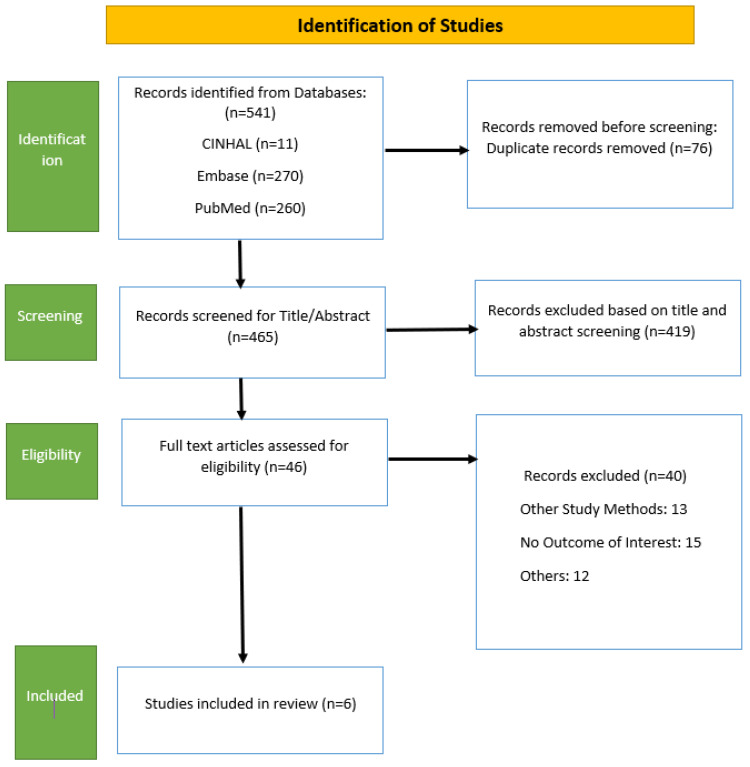
PRISMA flow diagram.

**Figure 2 healthcare-11-02635-f002:**
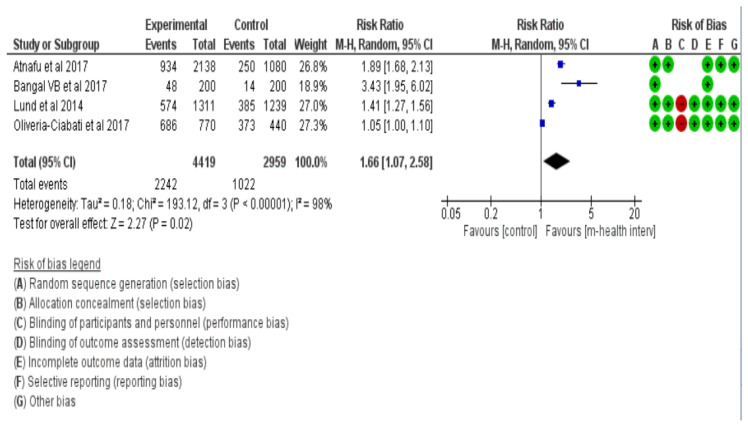
Forest plot for ANC attendance [25,26,27,28].

**Figure 3 healthcare-11-02635-f003:**
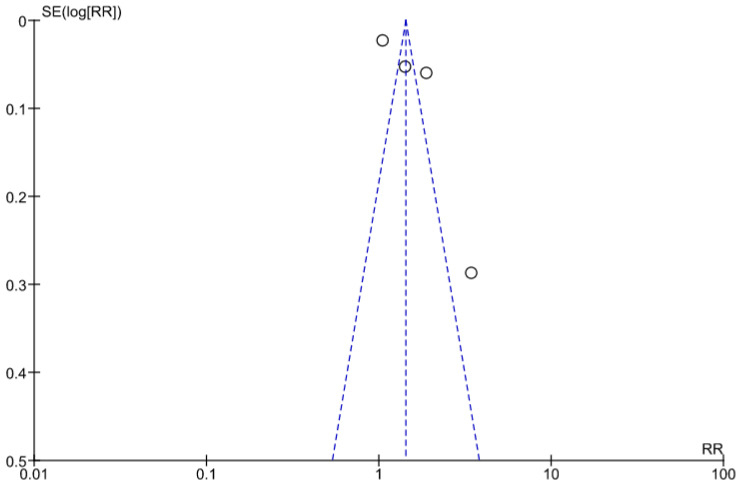
Funnel plot for ANC attendance.

**Table 1 healthcare-11-02635-t001:** The basic characteristics of all the individual studies have been outlined.

Study ID	General Information	Study Characteristics	Participant Characteristics	Intervention and Setting	Outcome Data/Results
Lund, 2014 [23]	**Date of data extraction** March 2009 to March 2010 **Article title** Mobile phones improve antenatal care attendance in Zanzibar: a cluster randomised controlled trial **Country of origin** Zanzibar, United Republic of Tanzania	**Aim/objectives of the study** To evaluate the association between a mobile phone intervention named “wired mothers” and antenatal care in Zanzibar **Study design** A pragmatic cluster-randomised controlled trial **Study inclusion and exclusion criteria** Women who attended antenatal care appointments at selected healthcare facilities. “Wired mothers” was used to describe women linked to the health system by use of a mobile phone intervention throughout their pregnancy and postpartum period **Recruitment procedures used (e.g., details of randomization and blinding)** Simple random allocation No masking	**Characteristics of the participants at the beginning of the study** 2550 pregnant women Interventions—131 Controls—1239	**Description of the intervention(s) and control(s)** Automated short messaging service (SMS) system	**ANC attendance** Four or more antenatal care visits Intervention—574/1311 (44%) Control—385/1239 (31%)
Omole et al., 2017 [24]	**Article title** The effect of mobile phone short message service on maternal health in south-west Nigeria **Country of origin** Nigeria	**Aim/objectives of the study** To assess the socio-economic and baseline maternal health characteristics of antenatal attendees To design and implement a maternal health SMS intervention among selected pregnant women **Study inclusion and exclusion criteria** **Recruitment procedures used (e.g., details of randomization and blinding)** A pregnant woman attending any of the study facilities must reside in the Ife-Ijesa zone, must own a mobile phone, and must be able to read and write in English or the Yoruba (local dialect) language	**Characteristics of the participants at the beginning of the study** 508 clients participated in the study, consisting of 248 in the control group and 260 in the intervention group	**Setting in which the intervention is delivered** 6 government-owned secondary health facilities 89 primary healthcare facilities managed by local government authorities 103 non-government facilities, including mission-owned health facilities	**Neonatal death** Control—67/248, 40.4% Intervention—65/260, 36.5% **ANC attendance** Improved ANC attendance—(99.4%)
Lund et al., 2014 [25]	**Article title** Mobile Phone Intervention Reduces Perinatal Mortality in Zanzibar: Secondary Outcomes of a Cluster Randomized Controlled Trial **Country of Origin** Zanzibar	**Aim/objectives of the study** To evaluate the association between a mobile phone intervention and perinatal mortality in a resource-limited setting. **Study design** Cluster randomised controlled trial **Study inclusion and exclusion criteria** **Recruitment procedures used (e.g., details of randomization and blinding)** Cluster sampling **Unit of allocation (e.g., participant, GP practice, etc.)** Primary healthcare facilities in Zanzibar	**Characteristics of the participants at the beginning of the study** 2550 pregnant women Interventions—1311 Controls—1239 **Total number**— 2550	**Description of the intervention(s) and control(s)** “Mobile Solutions Aiding Knowledge for Health Improvement” (M-SAKHI)	**Major adverse maternal outcome** Severe complications- Intervention—182/1311 (13.9%) Control—199/1239 (16.1%) **ANC attendance** Antenatal care (four or more visits) Intervention—574/1311 (43.8%) Control—385/1239 (31.1%) **Stillbirth** Intervention—22/54 Control—32/54 **Perinatal mortality** Intervention—25/69 Control—44/69
Oliveira-Ciabati et al., 2017 [26]	**Date of data extraction** April 2015 and March 2016 **Article title** PRENACEL—a mHealth messaging system to complement antenatal care: a cluster randomised trial **Country of Origin** Brazil	**Aim/objectives of the study** To determine whether the use of a bi-directional short message service (PRENACEL) providing information on pregnancy, childbirth, antenatal and intrapartum care, and able to answer the specific queries of pregnant women increases the coverage of recommended antenatal practices. **Study design** Cluster randomised trial	**Characteristics of the participants at the beginning of the study** 1210 pregnant women Interventions—770 Controls—440	**Setting in which the intervention is delivered** Primary health care units (PHCUs) **Description of the intervention(s) and control(s)** The PRENACEL SMS package was adapted from the Mobile Alliance for Maternal Action (MAMA)	**ANC practices** (≥6 antenatal visits) Intervention— Total—686/770 (89.1%) PRENACEL—112/116 (96.6%) Control—373/440 (84.8%)
Atnafu et al., 2017 [27]	**Date of data extraction** September 2012-October 2013 **Article title** The role of mHealth intervention on maternal and child health service delivery: findings from a randomised controlled field trial in rural Ethiopia **Country of Origin** Ethiopia	Aim/objectives of the study To assess the role of a mobile phone equipped with short message service (SMS)-based data-exchange software linking community health workers to health centers in rural Ethiopia and whether it affects selected MCH outcomes Study design Randomised control trial	**Characteristics of the participants at the beginning of the study** At baseline— Ezha (treatment 1)—1065 (98.6%) Abeshge (treatment 2)—1073 (99.35%) Sodo (control)—1080(100%) After intervention— Ezha (treatment 1)— 1066 (98.7%) Abeshge (treatment 2)—946 (87.6%) Sodo (control)— 1077 99.72%)	**Setting in which the intervention is delivered** **Description of the intervention(s) and control(s)** The FrontlineSMS-based application was offered only to the treatment groups, not the control groups.	**Number of ANC visits (≥4)**— At baseline— Ezha (treatment 1)—482/1065 (45.32%) Abeshge (treatment 2)—169/1073, (15.80%) Sodo (control)—264/1080, (24.48%) After intervention- Ezha (treatment 1)—637/1066, (59.84%) Abeshge (treatment 2)—297/946, (31.50%) Sodo (control)—250/1077 (23.27%)
Bangal VB et al., 2017 [28]	**Date of data extraction** September 2012-October 2013 **Article title** Use of mobile phone for improvement in maternal health: a randomised control trial **Country of Origin** India	**Aim/objectives of the study** To improve maternal health and pregnancy outcome through optimum utilisation of antenatal, natal, and postnatal care services, with the use of a mobile phone as a medium of communication between health care providers and communities in rural areas **Study design** Randomised control trial	**Characteristics of the participants at the beginning of the study** Total—400 Intervention—200 Control—200	**Setting in which the intervention is delivered** Rural Medical College, Loni, Ahmednagar **Description of the intervention(s) and control(s)** Mobile phone calls and text messages (SMS)— important aspects of antenatal care at regular intervals.	**Number of ANC visits** 5–6 ANC visits Control—33/200 (16.50%) Intervention—67/200 (33.50%) >6 ANC visits Control—14/200 (07.00%) Intervention—48/200 (24.00%)

**Table 2 healthcare-11-02635-t002:** Risk of bias of the included studies. Overall, five studies were assessed as low risk of bias, and one [28] as an unclear bias which is illustrated as below.

(**a**)		
**Bangal VB et al., 2017** [28]		
Methods	Randomised control trial	
Participants	400	
Interventions	Mobile phone calls, Text messages (SMS)	
Outcomes	ANC Attendance	
Notes		
**Risk of bias**		
**Bias**	**Authors’ judgment**	**Support for judgment**
Random sequence generation (selection bias)	Low risk	Randomly allocated to control and intervention group
Allocation concealment (selection bias)	Unclear risk	
Blinding of participants and personnel (performance bias)	Unclear risk	
Blinding of outcome assessment (detection bias)	Unclear risk	
Incomplete outcome data (attrition bias)	Low risk	
Selective reporting (reporting bias)	Unclear risk	
Other bias	Unclear risk	
(**b**)		
**Lund 2014** [23]		
Methods	Pragmatic cluster-randomised controlled trial	
Participants	2550	
Interventions	Automated short messaging service (SMS) system	
Outcomes	ANC Attendance	
Notes		
**Risk of bias**		
**Bias**	**Authors’ judgment**	**Support for judgment**
Random sequence generation (selection bias)	Low risk	Simple random allocation
Allocation concealment (selection bias)	Low risk	
Blinding of participants and personnel (performance bias)	High risk	Neither study participants nor clinic staff were masked
Blinding of outcome assessment (detection bias)	Low risk	
Incomplete outcome data (attrition bias)	Low risk	
Selective reporting (reporting bias)	Low risk	
Other bias	Low risk	
(**c**)		
**Lund et al., 2014** [25]		
Methods	Cluster randomised controlled trial	
Participants	2550	
Interventions	Mobile Solutions Aiding Knowledge for Health Improvement (M-SAKHI)	
Outcomes	ANC Attendance, still birth, maternal complication, perinatal mortality	
Notes		
**Risk of bias**		
**Bias**	**Authors’ judgment**	**Support for judgment**
Random sequence generation (selection bias)	Low risk	Simple random allocation
Allocation concealment (selection bias)	Low risk	
Blinding of participants and personnel (performance bias)	High risk	Cluster and study participant were not masked
Blinding of outcome assessment (detection bias)	Low risk	
Incomplete outcome data (attrition bias)	Low risk	
Selective reporting (reporting bias)	Low risk	
Other bias	Low risk	
(**d**)		
**Oliveria-Ciabati et al., 2017** [26]		
Methods	Cluster randomised trial	
Participants	1210	
Interventions	PRENACEL SMS package	
Outcomes	ANC Attendance	
Notes		
**Risk of bias**		
**Bias**	**Authors’ judgment**	**Support for judgment**
Random sequence generation (selection bias)	Low risk	Cluster randomisation
Allocation concealment (selection bias)	Low risk	
Blinding of participants and personnel (performance bias)	High risk	Participants and health professionals were not masked to the intervention
Blinding of outcome assessment (detection bias)	Low risk	
Incomplete outcome data (attrition bias)	Low risk	
Selective reporting (reporting bias)	Low risk	
Other bias	Low risk	
(**e**)		
**OMOLE et al., 2017** [24]		
Methods	Experimental	
Participants	508	
Interventions	Short message service	
Outcomes	Neonatal death	
Notes		
**Risk of bias**		
**Bias**	**Authors’ judgment**	**Support for judgment**
Random sequence generation (selection bias)	Low risk	
Allocation concealment (selection bias)	Low risk	
Blinding of participants and personnel (performance bias)	High risk	The research assistants and study subjects were not blinded
Blinding of outcome assessment (detection bias)	Low risk	
Incomplete outcome data (attrition bias)	Low risk	
Selective reporting (reporting bias)	Low risk	
Other bias	Low risk	
(**f**)		
**Atnafu et al., 2017** [27]		
Methods	Randomized control trial	
Participants	3218	
Interventions	Frontline SMS-based application	
Outcomes	ANC Attendance	
Notes		
**Risk of bias**		
**Bias**	**Authors’ judgment**	**Support for judgment**
Random sequence generation (selection bias)	Low risk	Randomly assigned by lottery method
Allocation concealment (selection bias)	Low risk	Random allocation
Blinding of participants and personnel (performance bias)	Unclear risk	
Blinding of outcome assessment (detection bias)	Unclear risk	
Incomplete outcome data (attrition bias)	Low risk	Data is recorded for all participants
Selective reporting (reporting bias)	Low risk	Prespecified outcome was reported
Other bias	Low risk	

## Data Availability

The authors declare that the data collected was gathered from publicly available databases and is available with this publication.

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
