# Peer review of "Effectiveness of mHealth Interventions for Monitoring Antenatal Care among Pregnant Women in Low- and Middle-Income Countries: A Systematic Review and Meta-Analysis"

_healthcare, 2023, doi:10.3390/healthcare11192635_

Round 1

Reviewer 1 Report

In the current study, the authors undertake a systematic review and meta-analysis of trials to assess the efficacy of mHealth treatments for monitoring antenatal health among pregnant women in LMICs. They demonstrated that mHealth technologies are increasingly being employed to improve care accessibility and maternal and fetal health, and that their use might greatly enhance antenatal care provision.

However, I have some suggestions.

1.    The manuscript should be properly proofread for minor typographical, syntactical, and grammatical errors.

2.    Abbreviations should be defined the first time they are used in the abstract and the main body.

3.    Add definitions of low-income and middle-income countries.

4.    The type of studies is not clear, please add inclusion and exclusion criteria for studies included.

5.    More specificity is required for types of participants.

6.    The section “dentification and characteristics of included studies for SR and MA” needs more clarification of the study number and selection e.g., total studies collected were 541 after removing duplicate 73 studies the remaining study was 419, how? Also, the PICO question needs more explanation. What are the reasons for excluding the other 40 papers?

7.    Authors must adhere to the journal's format style.

The manuscript should be properly proofread for minor typographical, syntactical, and grammatical errors.

Author Response

Thank you for your suggestions.

Reviewer 2 Report

Dear author,

Author Response

Thank you for your suggestions.
